# Hybrid Bis-Histidine Phenanthroline-Based Ligands to Lessen Aβ-Bound Cu ROS Production: An Illustration of Cu(I) Significance

**DOI:** 10.3390/molecules26247630

**Published:** 2021-12-16

**Authors:** Marielle Drommi, Clément Rulmont, Charlène Esmieu, Christelle Hureau

**Affiliations:** CNRS, LCC (Laboratoire de Chimie de Coordination), 205 Route de Narbonne, CEDEX 4, 31077 Toulouse, France; marielle.drommi@lcc-toulouse.fr (M.D.); clement.rulmont@lcc-toulouse.fr (C.R.); charlene.esmieu@lcc-toulouse.fr (C.E.)

**Keywords:** Alzheimer’s disease, copper, amyloid-β, ROS production, peptide, phenanthroline

## Abstract

We here report the synthesis of three new hybrid ligands built around the phenanthroline scaffold and encompassing two histidine-like moieties: phenHH, phenHGH and H’phenH’, where H correspond to histidine and H’ to histamine. These ligands were designed to capture Cu(I/II) from the amyloid-β peptide and to prevent the formation of reactive oxygen species produced by amyloid-β bound copper in presence of physiological reductant (e.g., ascorbate) and dioxygen. The amyloid-β peptide is a well-known key player in Alzheimer’s disease, a debilitating and devasting neurological disorder the mankind has to fight against. The Cu-Aβ complex does participate in the oxidative stress observed in the disease, due to the redox ability of the Cu(I/II) ions. The complete characterization of the copper complexes made with phenHH, phenHGH and H’phenH’ is reported, along with the ability of ligands to remove Cu from Aβ, and to prevent the formation of reactive oxygen species catalyzed by Cu and Cu-Aβ, including in presence of zinc, the second metal ions important in the etiology of Alzheimer’s disease. The importance of the reduced state of copper, Cu(I), in the prevention and arrest of ROS is mechanistically described with the help of cyclic voltammetry experiments.

## 1. Introduction

Alzheimer’s disease (AD) is the most common form of senile dementia, causing loss of autonomy, and affecting around 50 million people worldwide, mainly people over 60 [1]. Since the world population is aging, the social and economic consequences of this disease are expected to increase: related costs are to soar up to 2 trillion US$ by 2030 [2]. This neurodegenerative disease is characterized by different pathological markers, including extracellular deposits of amyloid-β peptide (Aβ) aggregates denominated as senile plaques, metal ions dyshomeostasis and an overall oxidative stress [3,4,5,6,7,8,9,10]. Aβ is a 40 to 42 amino-acid residues peptide, made of a hydrophilic N-terminus, with amino-acid residues able to bind metal ions, and a hydrophobic C-terminus part prone to aggregate (Figure 1a). The interaction between Aβ and metal ions has been extensively studied, and it has been shown that Aβ can bind metal ions, especially Cu, Zn and Fe ions [6,11,12,13,14,15,16,17,18,19,20]. In particular, copper is a redox-active metal, able to cycle between two biologically relevant redox states: Cu(I) and Cu(II). Cu(II) can be reduced by various reductants (e.g., ascorbate) and Cu(I) can be incompletely oxidized by dioxygen in stepwise single-electron transfer, leading to the formation of reactive oxygen species (ROS), such as O_2_^•−^, HO^•^, H_2_O_2_ and hence participating to oxidative stress [3,7,21,22,23]. In a healthy brain, their concentrations are strictly regulated by dedicated enzymes, including Cu enzymes in which the environment (and thus, the redox state) of Cu is tightly controlled [24,25]. However, in AD, brain tissues are damaged by oxidative stress, in link to the high-concentration of “loosely-bound” Cu in Aβ aggregates (Figure 1b) [25].

To address the deleterious effects of Cu-bound Aβ, a large number of ligands has been developed. These ligands should combine many properties to be considered as potential therapeutic molecules [15,26,27,28]. Thermodynamically, they should be able to remove Cu from Aβ, which means that they must have a higher affinity for Cu(II) or Cu(I) than Aβ (10^10^ M^−1^ at pH 7.4 [14,29] or 10^7^ M^−1^–10^10^ M^−1^ [30,31,32], respectively), but this affinity should not be too high, to avoid removing Cu from essential metalloenzymes and disrupt Cu homeostasis [24,33]. The complex formed by the ligand with Cu should be stable in its redox state, Cu(I) or Cu(II), not to form ROS on its own. The ligand should also be selective for Cu against Zn [34], especially since Zn is more abundant than Cu in the synaptic cleft (1–10 µM for Cu and 10–300 µM for Zn under neuronal excitation [10,35,36]). Furthermore, the kinetics of complex formation between the ligand and Cu has to be fast, to prevent efficiently ROS formation [37]. Finally, the ligand should respect the criteria linked to therapeutic applications: it should be stable *in vivo* (especially at different pH), fulfill pharmacokinetics properties, and be able to cross the brain blood barrier [27].

One particular ligand has drawn the attention on the fact that it is important to consider both Cu(II) and Cu(I) chelation to the ligand and the respective speciation [38]. This ligand, named phenH, is composed of a peptide-like scaffold, with a histidine linked to a phenanthroline in position 2, and an amide at the C-terminus. This ligand is able to chelate Cu(II) in a distorted square-planar coordination, with the phenanthroline, the deprotonated amide of the peptide bond and the imidazole ring [39]. When added in a 1:1 ratio against Cu, phenH is able to slow ROS production, in absence and in presence of Aβ, being a good candidate as chelator. However, strikingly, when phenH is added at higher ratio (2 equivalents against Cu), its ability to slow ROS production is lowered: usually, the more ligand is added, the slower is the ROS production, because there is less redox active loosely-bound copper [40]. To explain this unexpected phenomenon, the following mechanism has been proposed: when phenH is in excess, a Cu(I)(phenH)_2_ complex is formed during the redox cycling, with Cu(I) being chelated by two phenanthroline moieties in a tetrahedral fashion (Figure 1b). This 1:2 complex is redox active, and thus is less efficient in preventing ROS production. This effect is alleviated by the addition of Zn(II), because association constant of Zn(II)(phenH) is lower than the one of Cu(II)(phenH) but higher than the one of Cu(I)(phenH)_2_, so Zn(II) can take over the excess of ligand, preventing the formation of the 1:2 complex [38]. 

The aim of the present work was to optimize phenH, with new generation of phenanthroline-based peptide scaffolds, that would less be able to create this 1:2 complex, and thus could avoid ROS production even in excess of ligand. To enlarge the family of phenanthroline-based ligands, we designed the set of molecules presented below (Figure 2). PhenHH (*resp.* phenHGH) is made of a phenanthroline moiety appended with a peptide sequence of two adjacent histidines (*resp.* histidine-glycine-histidine fragment). Finally, H’phenH’ is a symmetrical version of phenH, but with histamines instead of histidines, for synthetic reasons. PhenH was kept in the set of studied molecules as a reference. These ligands were rationally designed following two hypothesis:Increasing the sterical hindrance of the side chains of the phenanthroline would prevent the formation of the Cu(I)(phen)_2_ complex, responsible for ROS production.The addition of the second imidazole moiety would allow the formation of a 1:1 tetrahedral Cu(I) complex, with the metal being bound by the phenanthroline and both imidazole moiety, and this complex would be more thermodynamically favorable than Cu(I)(phen)_2_ complex.

Herein, we report the complete characterization of the Cu(II) complexes made with these ligands, their ability to stop ROS production, and an electrochemical study of the ligands in the presence of Cu.

## 2. Results and Discussion

### 2.1. Synthesis of the Ligands

Asymmetrical ligands phenH, phenHH and phenHGH were synthesized from their carboxylic acid precursor (Appendix A): 2-carboxy-1,10-phenanthroline, which was synthesized according to a described procedure [41]. 1,10-phenanthroline was activated as an N-oxide through oxidation by hydrogen peroxide in acidic conditions. A nucleophilic aromatic substitution in presence of cyanide and benzoyl chloride led to 2-cyano-1,10-phenanthroline. Finally, the hydrolysis of the nitrile in basic medium led to 2-carboxy-1,10-phenanthroline.

To yield the ligands phenH, phenHH and phenHGH, the amino-acids chain was first grafted to a rink amide resin *via* solid phase peptide synthesis (SPPS) standard protocol [42]. Briefly, fluorenylmethoxycarbonyl (Fmoc)-protected resin was deprotected using 20% piperidine in dimethylformamide (DMF), and the loading of the resin was calculated by UV-vis titration of the dibenzofulvene-piperidine adduct [43]. Fmoc-protected amino acids were coupled using hydroxybenzothiazole (HOBt) and (2-(1*H*-benzotriazol-1-yl)-1,1,3,3-tetramethyluronium hexafluorophosphate (HBTU) as activators and diisopropylethylamine (DIPEA) as base. 2-carboxy-1,10-phenanthroline was then coupled using benzotriazol-1-yloxytripyrrolidinophosphonium hexafluorophosphate (PyBOP) as activator, which gave better yields than HBTU [39]. Finally, the ligand was cleaved from the resin in acidic medium, and purified by reverse phase chromatography. 

Symmetrical ligand H’phenH’ was synthesized from the dicarboxylic acid precursor: 2,9-dicarboxy-1,10-phenanthroline, which was synthesized according to a described protocol [44]. 2,9-dimethyl-1,10-phenanthroline was oxidized to the dialdehyde with selenium dioxide, and then oxidized again to the dicarboxylic acid in nitric acid. It was first tried to couple the diacid precursor to two histidines at the same time, using SPPS, but probably because of the distance between two histidines grafted on the resin, this method gave very low yields (approx. 2%). The solution phase peptide coupling strategy was thus adopted (57% yield), and histamine was used instead of histidine as the amine, for synthesis simplicity reasons. Ligand H’phenH’ was synthesized according to a described protocol [45]: treatment of 2,9-dicarboxy-1,10-phenanthroline with carbonyldiimidazole (CDI) led to the corresponding activated ester, to which was added histamine to yield H’phenH’.

### 2.2. Speciation of the Cu(II) Complexes

To determine the structures and speciation of the Cu(II) complexes, UV-visible and electron paramagnetic resonance (EPR) spectroscopies were used. 

-Firstly, the ligands were studied in the presence of 1 equivalent of Cu(II), to determine the structure of the complexes Cu(II)L (L = phenH, phenHH, phenHGH and H’phenH’ unless otherwise stated). For phenH, phenHH and phenHGH, the UV-visible spectra overlap (Figure 1a), the three complexes have comparable *λ*_max_^d-d^ and *ε* values (Table 1). Furthermore, their EPR signatures are overlapping as well: they have same values of the characteristic EPR g-factors and hyperfine coupling constants. This indicates that the coordination sphere of Cu(II) inside phenHH and phenHGH is virtually identical to that in phenH. The latter was determined previously with its crystal structure [38]. A coordination mode for Cu(II)(phenHH) and Cu(II)(phenHGH) could thus be proposed on this basis (Appendix A). For H’phenH’, it can be noticed that the EPR signature of Cu(II)(H’phenH’) is significantly different from the signature of Cu(II)(phenH) as mainly mirrored by different A_//_ values (Table 1). In addition, the d-d transition band is redshifted: the coordination sphere of Cu(II) inside H’phenH’ is thus different from the one in phenH, phenHH and phenHGH. The redshift of the d-d transition band suggests the decoordination of the deprotonated amide ligand [46], while the *ε* value indicates that mainly nitrogen atoms are bound to the Cu(II). The EPR parameters values are also in line with a 4N coordination, according to Peisach-Blumberg correlation [47,48]. On this basis, a structure for Cu(II)(H’phenH’) complex has been proposed (Appendix A), where the metal ion is chelated by the phenanthroline and both imidazole rings.-In a second time, the possibility to form the Cu(II)L_2_ complex was investigated by EPR spectroscopy, for phenHH, phenHGH and H’phenH’. EPR spectra of Cu(II) in presence of 1 and 2 equivalents of ligands were monitored (Appendix A). The EPR signatures of Cu(II) in these two conditions are the same, for all the ligands: the Cu(II)L_2_ species are not detected, in line with the previously published data [38].-Finally, the behavior of the ligands in the presence of more equivalents of Cu(II) was investigated. EPR spectra of phenHH in presence of 2 equivalents of Cu(II) has been performed (Appendix A), and shows the superimposition of a remaining contribution from the signature of the 1:1 species and a broad and weak signal (better seen with modifying the recording conditions, spectrum noted with a (*)). This suggests the formation of a binuclear copper complex, with two close (<7 Å) paramagnetic centers, leading to the disappearance of EPR signal due to magnetic interactions [24]. In UV-visible spectroscopy, a weak redshift in the d-d absorption wavelength is observed as well as a 30% increase in the absorbance in presence of 2 equivalents of Cu(II) (Appendix A, Appendix A). This indicates that the second Cu(II) ion may be bound by water molecule as well as the two remaining nitrogen-donor atoms of phenHH [46,49]. On this basis, different structures have been proposed, such as Cu_2_(phenHH), with a second copper being likely bound between the imidazole of the second histidine and the deprotonated amide of the peptide bond [50,51] (Appendix A). Other structures have been considered as well for intermediate Cu:phenHH ratio, such as a Cu_3_(phenHH)_2_ complex (Appendix A). These suggestion are in line with the observation made during the titrations of the ligands by Cu(II) (Appendix A). In fact for phenHH, two slope breaks were observed, suggesting the formation of species with stoichiometry in Cu(II) higher than 1:1.

For phenHGH and H’phenH’, similar observations can be made (Appendix A, Appendix A): hypothetical structures for Cu(II)_2_(phenHGH), Cu(II)_3_(phenHGH)_2_ and Cu(II)_2_(H’phenH’) have been proposed on this basis (Appendix A). 

On the contrary for phenH, the EPR spectra performed in presence of 2 equivalents of Cu(II) does not show any weakening of the signal (Appendix A), suggesting that no binuclear complex is formed with phenH in excess of copper. Furthermore, the UV-visible spectra of phenH with 2 equivalents of copper does not show any shift of the d-d absorption wavelength, but the precipitation of Cu(II) (overall absorbance increase, Appendix A), due to the poor solubility of free Cu(II) in HEPES buffer.

### 2.3. Ability of the Ligands to Extract Cu(II) from Aβ

The ligands were then tested for their ability to extract Cu(II) from the Aβ peptide. EPR spectra of Cu(II) in the presence of 1 equivalent of the ligands and 1 equivalent of Aβ were realized, with and without Zn(II).

For phenHH and phenHGH (Appendix A), the EPR signature of Cu(II) in presence of the ligands, Aβ and with or without Zn(II) is identical to the one of Cu(II) in presence of the ligands. This means that the affinity of Cu(II) for the ligands is higher than the one of Cu(II) for Aβ, and that the ligand is selective for Cu(II) vs. Zn(II) in presence of Aβ.

On the contrary for H’phenH’ (Appendix A), two species can be seen in the EPR spectra in the presence of H’phenH’ and Aβ: a majority of Cu(II)(H’phenH’) and a minority of Cu(II)Aβ (in an approximatively 7/3 ratio). The affinity constants of Cu(II)(H’phenH’) and Cu(II)Aβ are thus in the same order of magnitude. The addition of Zn(II) does not displace this equilibrium. It is then anticipated that ROS production will less be slowed by H’phenH’ in the presence of Aβ. 

### 2.4. Ability of the Ligands to Stop Cu-Induced ROS Production

In order to study and quantify the copper-induced ROS production, an experiment based on ascorbate consumption has been used [52,53]. Indeed, ascorbate concentration can be measured by UV-visible spectroscopy by monitoring absorbance at 265 nm (*ε* = 14,500 M^−1^ cm^−1^). This is a straightforward method, which compares well with fluorescence-based ROS detection method [14,52,54,55,56]. In presence of copper, dioxygen, ascorbate and other species (such as Aβ peptide or potential drug candidates), the ascorbate consumption rate will provide information on ROS production rate, and thus on oxidative stress produced in this environment. 

The ligands have thus been challenged to stop ROS production, in absence and in presence of Aβ and/or Zn(II). First, the global behavior of ligand phenHH, phenHGH and H’phenH’ is similar to the one of phenH (Figure 2 and Appendix A): in absence of Zn(II) (without or with Aβ), the ligands slow down the Cu-induced ascorbate consumption more efficiently when added at 1.2 equivalent than 2 equivalents, suggesting that the mechanism described before applies here as well (i.e., formation of a Cu(I)L_2_ complex able to redox cycle, Figure 1b). The addition of Zn(II) prevents the formation of the Cu(I)L_2_ complex by competing for ligand binding and thus give back the ligands the ability to lessen ROS production. These results show that the crowding brought by the His-His, His-Gly-His or bis-histamine moieties is not enough to preclude formation of the redox active Cu(I)L_2_ complex. 

It is however interesting to compare the different ligands side by side to evaluate their performance. First concentrating on the complex CuL in a 1:1 ratio (Figure 2c, dark blue bars), ligands phenH, phenHH and phenHGH exhibit similar ascorbate consumption rates (4.4 ± 0.2 nM s^−1^ for phenH, 4.91 ± 0.04 nM s^−1^ for phenHH and 5.4 ± 0.6 nM s^−1^ for phenHGH). This is in line with the similarity of the coordination sphere of Cu(II) in these ligands, as shown earlier (Figure 1b and Appendix A)): Cu(II) is tightly bound in a square-planar coordination, unable to cycle back to Cu(I) and produce ROS. In contrast, for H’phenH’ the ascorbate consumption is much faster (11.6 ± 0.3 nM s^−1^): this may be explained by the coordination sphere of H’phenH’ around Cu(II) (Appendix A), where the metal ion is more loosely bound (with three adjacent metallacycles of (9, 5, 9) members vs. (5, 5, 6) for phenH, phenHH and phenHGH [50]). This more flexible environment may provide Cu(I)/Cu(II) redox active species. This trend is also observed regardless the Cu:L stoichiometry. In presence of Aβ, the ascorbate consumption rate is also higher than for the other three ligands; this may be due to the only partial removal of Cu(II) from Aβ by H’phenH’. 

In the presence of Zn(II), the ascorbate consumption rate is similar for all four ligands. For phenH, phenHH and phenHGH, it is not very different from the rate in absence of Zn(II) with 1 equivalent of ligand, showing that these 3 ligands are selective for Cu(II). However for H’phenH’, the addition of Zn(II) divides the rate by more than a factor 2 (11.6 ± 0.3 nM s^−1^ without Zn(II), 4.2 ± 0.3 nM s^−1^ with Zn(II)). To explain the positive effect of Zn(II) in this situation, it is hypothesized that H’phenH’ has a similar affinity for Zn(II) than for Cu(II), and thus binds to Zn(II), in part. As demonstrated previously, H’phenH’ is able to bind two Cu(II) ions (N from the phenanthroline, from the amide and from the imidazole, Appendix A). We thus propose the following reaction:2 Cu(II)(H’phenH’) + Zn(II) ⇄ Cu(II)_2_(H’phenH’) + Zn(II)(H’phenH’)

This hypothesis is supported by the EPR spectra of Cu(II) in the presence of H’phenH’ and Zn(II) (Appendix A): when adding 1 equivalent of Zn(II) to a solution of Cu(H’phenH’), the EPR spectra is the overlay of the spectra of Cu(II)(H’phenH’) at approximately 65%, and of a broad spectra, similar to the one obtained for Cu(II)_2_(H’phenH’) (Appendix A). In the Cu(II)_2_(H’phenH’) coordination mode, Cu(II) may be less able to cycle back to Cu(I) (due to coordination of deprotonated amide), and thus produces less ROS.

### 2.5. Electrochemical Exploration of the Mechanism

As phenH remains the most efficient ligand to slow down the ROS production induced by Cu(II), we decide to bring supplementary evidence to the mechanism presented in the introduction, using cyclic voltammetry experiments. Potentials are all given vs. SCE.

To study the redox activity of copper, the scanned potentials ranged from 0.5 V to −0.8 V and back to 0.5 V. The redox activity of phenH alone was first investigated: when scanning from 0.2 V to −1 V and then to 1 V, neither reduction nor oxidation wave can be seen: the cyclic voltammogram (CV) can be overlapped to the blank. PhenH has thus no redox activity in the potential window where the copper activity will be probed afterwards.

In a first experiment, a study at a single scan rate (100 mV/s) was carried out, where the ratio of phenH vs. Cu(II) (Figure 3a) was varied. In presence of 1 equivalent of phenH at pH 7.1, the CV is characteristic of a quasi-reversible system, with E_p_^1/2^ = −0.30 V and ΔE_p_ = 0.16 V (E1). When the phenH to Cu ratio increases, the anodic wave at −0.4 V is not modified, but the cathodic wave observed near −0.3 V disappears while another one appears at 0.05 V. This phenomenon could witness a so-called Chemical-Electrochemical mechanism where chemical reaction would be an equilibrium between the electro-generated Cu(I)(phenH) complex with a Cu(I)(phenH)_2_ complex (C1), the first one being oxidized at −0.3 V (E1), and the second one at 0.05 V (E2). This latter values is similar to that reported for the [Cu(phen)_2_]^2+^ complex [57].

Cu(II)LH_−1_ + e^−^ + H^+^ ⇄ Cu(I)L(E1)LH_−1_ = L with deprotonated amideCu(I)L + L’ + (H^+^) ⇄ Cu(I)L’_2_(C1)L’ = L or LH^+^ (with protonated His)Cu(I)L’_2_ → Cu(II)L’_2_ + e^−^(E2)
Cu(II)L’_2_ → Cu(II)LH_−1_ + L’ + H^+^ + (H^+^)(C2)


The effect of ligand excess is abolished by the addition of Zn(II) in link with recruitment of excess ligand to form the Zn(II) complex as described in the introduction (Figure 3a green). On the CVs in presence of Zn(II), the sharp oxidation peak may come from the oxidation of solid Cu(0) that deposits at the surface of the electrode when scanning at the low potentials, indicating that Zn(II) can shift weak amount of Cu(I) outside the ligand.

To further explore the kinetics of the CE mechanism reactions, Cu(II) was then studied in presence of 1 and 2 equivalents of phenH while varying the scan rate at pH 7.1 (Figure 3d,e). Strikingly with 1 equivalent of ligand, when the scan rate increases, the oxidation wave at 0.05 V appears, weak at a scan rate of 200 mV/s, but predominant at 500 mV/s. We propose that this equilibrium (reaction C1) is extremely quickly achieved and shifted to the formation of the Cu(I)(phenH)_2_ species. When the CV is performed at a scan rate of 500 mV/s, it scans the system in a “frozen state” where Cu(I)(phenH)_2_ species are predominant, and thus only the oxidation wave at 0.05 V can be seen. When the scan rate decreases, when the potential reaches −0.3 V, the small proportion of Cu(I)(phenH) in solution is oxidized (so consumed), which shifts the equilibrium towards the formation of more Cu(I)(phenH), which still has the time to be oxidized. If the scan rate is slow enough (below 100 mV/s), all Cu(I)(phenH)_2_ is thus consumed before reaching 0.05 V, and the oxidation wave at 0.05 V cannot be seen.

With 2 equivalents of ligand (Figure 3e), no difference could be seen on the different CVs while varying the scan rate (except the one due to the change of scan rate: higher intensity and shift of the waves). The equilibrium (C1) is shifted towards the formation of Cu(I)(phenH)_2_, so only the oxidation wave at 0.05 V is observed.

An important effect of the pH can be noticed on the CV with 1 and 2 equivalents of phenH (Figure 3b,c). At 1 equivalent of phenH, when increasing the pH to 8.5, the CV is characteristic of a reversible system, with E_p_^1/2^ = −0.30 V and ΔE_p_ = 0.16 V, whereas when decreasing the pH to 5.5, the CV exhibits the oxidation wave at −0.05 V, characteristic of the oxidation of Cu(I)(phenH)_2_. This can be explained by the state of protonation of the histidine during reaction (C1): in Cu(I)(phenH), the histidine must be deprotonated to form the complex, whereas in Cu(I)(phenH)_2_, it can be either protonated or deprotonated (Figure 3). A more acidic medium thus shifts this equilibrium towards the formation of Cu(I)(phenH)_2_, and so the anodic wave at 0.05 V is observed, whereas a more basic medium shifts the equilibrium towards the formation of Cu(I)(phenH), thus showing the reversible process at −0.3 V. In presence of 2 equivalent of phenH, a similar trend is observed but to a lesser extent, in line with the shift of the equilibrium C1 towards Cu(I)(phenH)_2_ in presence of 2 equivalent of ligand. 

Previous electrochemical experiments led on phenH [39] are in agreement with the CVs obtained in this study. At pH 7.4 and with a 1:1 ratio of Cu vs. phenH, a reduction wave at approximately −0.4 V and an oxidation wave at 0.1 V were observed, corresponding to the oxidation of Cu(I)(phenH)_2_. The oxidation wave of Cu(I)(phenH) is not observed probably because of the different buffering conditions used in this study.

Cyclic voltammetry experiments in presence of the other ligands have also been conducted, at a 1:1 ratio of copper against the ligand (Appendix A). For phenHH and phenHGH, it is first noticeable that the reduction wave of Cu(II) is broader than that of phenH and happens between −0.3 V and −0.6 V. This could be explained by the presence in solution of different Cu(II) species due to different speciation states (cf. Section 2.2) or protonation states (because of the second imidazole). These different states are in equilibrium one with another and have different reduction potentials. The oxidation potential of the electrogenerated Cu(I) species is at 0.1 V, and the wave profile is very similar to the one encountered for the oxidation of the Cu(I)(phenH)_2_ species (Figure 3a). The equilibrium (C) would thus be even more shifted towards Cu(I)L_2_ with the ligands phenHH and phenHGH. This is in line with the ROS experiments results: indeed, Cu-induced ROS production was faster in presence of phenHH or phenHGH than phenH, indicating the predominance of Cu(I)L_2_. 

Due to the poor solubility of H’phenH’ in water, this ligand was studied at 0.1 mM with 1% methanol (instead of 0.5 mM) (Appendix A). The faradic contribution to the cyclic voltammograms is thus less important than with the previous ligands, and the signal/noise ratio is worse. The reduction wave of Cu(II) is observed at −0.2 V: this higher value compared to previously described ligand could be due to the weaker stabilization of Cu(II) in water induced by H’phenH’ than that of phenH, because the metal is more loosely bound to the ligand and without participation of amide bond in its coordination (cf. Section 2.2). Similar to the other bis-histidine ligands, the process is not reversible, and the oxidation wave occurs at 0.2 V. This higher anodic potential compared to phenH, phenHH and phenHGH could indicate that the complex Cu(I)L_2_ is more stable with L = H’phenH’ than with L = phenHH or phenHGH. This is due to a weaker ability to generate the Cu(I) intermediate species, since the imidazole ring is farer (5-metallacycle vs. 8-metallacycle), and thus the C1 reaction is more shifted towards the Cu(I)(H’phenH’)_2_ species. The redox behavior is thus in line with the higher ascorbate consumption rate measured for H’phenH’.

## 3. Conclusions

In the present work, we have studied a wide family of phenanthroline-based peptide ligands in order to challenge the possibility of modifying the speciation at the Cu(I) level compared to a reference ligand, and thus improve its ability to lessen Cu and Cu-Aβ induced ROS production, in the context of AD. First, ligands phenHH, phenHGH and H’phenH’ exhibited interesting Cu(II) coordination properties, especially in presence of more than one equivalent Cu(II). The formation of complexes with more than 1 Cu(II) for 1 ligand is speculated, for instance Cu(II)_2_L or Cu(II)_3_L_2_, which is not observed with the parent phenH ligand. Second, with respect to Cu(I) speciation during the ROS production, the new ligands compare with the previously studied phenH, as probed by the monitoring of Cu and Cu-Aβ ROS production at 1.2 and 2 equivalents of ligands. This indicates that neither the crowding of phen moiety nor the availability of two His groups to complete a tetrahedral coordination is sufficient to prevent the formation of Cu(I)L_2_. Other synthetic modifications are currently investigated to tackle this issue.

Lastly, the mechanism of redox cycling has been deeply probed by cyclic voltammetry for phenH in presence of Cu and Zn, evidencing an equilibrium between the Cu(I)(phenH) or Cu(I)(phenH)_2_ species depending on the conditions (ratio Cu:ligand, presence of Zn, pH, scan rate). The other ligands have as well been studied, and in line with ROS data, the electrochemical study shows the predominance of the Cu(I)L_2_ species. Hence, the electrochemically-based mechanism perfectly matches the results obtained in the ROS production study.

## 4. Materials and Methods

### 4.1. Materials

All chemicals were purchased from Sigma-Aldrich, TCI Chemicals and Fluorochem and used without purification.

Aβ_16_ peptide (DAEFRHDSGYEVHHQK) was bought from Genecust.

Ligands phenH, phenHH, phenHGH and H’phenH’ were synthesized according to the protocols presented in the Appendix A.

### 4.2. Stock Solutions

All the stock solutions (except H’phenH’) were prepared in Milli-Q water (resistivity: 18.2 MΩ.cm).

Sodium salt of 2-[4-(2-hydroxyethyl)piperazin-a-yl] ethanesulfonic (HEPES) acid buffer was prepared at an initial concentration of 500 mM, and its pH was adjusted at 7.1 with 5 M sodium hydroxide solution.

Cu(II) stock solutions were prepared from CuSO_4_, 5 H_2_O at 100 mM and were titrated using UV-vis spectroscopy (*ε*_800 nm_ = 12 M^−1^ cm^−1^).

Zn(II) stock solutions were prepared were prepared from ZnSO_4_, H_2_O at 100 mM.

Aβ stock solutions were prepared at 10 mM and were titrated with UV-vis spectroscopy using the tyrosine chromophore (*ε*_276 nm_ = 1410 M^−1^ cm^−1^ in acidic conditions).

Ascorbate solutions were freshly prepared every week from sodium ascorbate at 5 mM.

PhenH, phenHH and phenHGH stock solutions were prepared at 10 mM and their precise concentrations were determined at first using Cu(II) titration with a solution of known concentration using the ligand absorption of the complex. Molar absorption coefficients of the ligands were then measured, allowing a simpler titration of the ligand stock solutions. H’phenH’ stock solutions were prepared in a similar way as the other ligands, but using methanol as solvent.

### 4.3. Ascorbate Consumption Experiments

UV-vis kinetic data of ascorbate consumption experiments were recorded with a Hewlett Packard Agilent 8453 spectrophotometer at a controlled temperature of 25 °C in a 1 cm path length quartz cuvette, with 500 rpm stirring.

The absorbance at 265 nm (ascorbate absorption band) was monitored every 30 s, corrected by the absorbance at 800 nm, and plotted as a function of time. The solutions were prepared in situ from stock solutions of Cu(II), Zn(II), Aβ and the ligands at 1 mM, and diluted to 12 µM for Zn(II) and Aβ, 12 or 24 µM for the ligands, and 10 µM for Cu(II), in 100 mM HEPES buffer (pH 7.1). Ascorbate was freshly prepared and diluted to 100 µM. The final volume in the cuvette was adjusted to 2 mL with Milli-Q water.

Before each experiment, a blank with 100 mM HEPES buffer was performed. Ascorbate was first added in the cuvette, 30 s later Cu(II) (and in some experiments Zn(II) and/or Aβ) was added. When the absorbance reached 1.1, ligand was added.

Ascorbate consumption rate were calculated between 2000 and 3000 s, by dividing the slope of the variation of absorbance by the extinction coefficient of ascorbate, *ε*_265 nm_ = 14,500 M^−1^ cm^−1^ [40]. Measurements were performed on at least two independent measurements.

### 4.4. Electron Paramagnetic Resonance Spectroscopy

Electron paramagnetic resonance (EPR) spectra were recorded using an Elexsys E-500 Bruker spectrometer, operating at a microwave frequency of approximatively 9.5 GHz. Spectra were recorded using a microwave power of 5 mW, with a magnetic field range of 2400 to 3700 G and an amplitude modulation of 5 G, and with an attenuation of 16 dB. Experiments were carried out at 117 K using a liquid nitrogen cryostat.

EPR samples were prepared in Eppendorf tubes from a 10 mM Cu(II) stock solution, diluted to 500 µM, in 50 mM HEPES buffer (pH 7.4). Various equivalents of ligands (0.5 to 2) were added from 10 mM stock solutions. As a cryoprotectant, 10% glycerol was added. The final volume was adjusted to 200 µL with milli-Q water. The mixture was transferred in an EPR quartz tube and frozen in liquid nitrogen.

A blank spectrum with 50 mM HEPES buffer was also recorded and subtracted from the other samples’ spectra.

### 4.5. Cyclic Voltammetry

Cyclic voltammetry (CV) experiments were performed using an Autolab PGSTAT302N potentiostat controlled with EC-Lab software. A three-electrode setup was used, consisting of a glassy carbon disk (3 mm in diameter), a saturated calomel electrode as reference electrode and a platinum wire as auxiliary electrode, in an argon-flushed 2 mL cell. The working electrode was carefully polished before each measurement on a red disk NAP (Struers) with 1 μm AP-A suspension first, 0.3 µM AP-A suspension then, during at least 5 min between each measurement. The solution was degassed during at least 5 min with argon between each measurement.

The scan rate was 100 mV/s unless indicated otherwise in the Figure’s caption. Support electrolyte is the HEPES buffer. 3 scans were realized for each experiment, and unless stated otherwise, it is the first scan which is shown on the figures. 

Before each experiment, a blank with 50 mM HEPES (pH 7.1) buffer was performed on a small range and on a large range of potential.

## Data Availability

The date are available on request from the corresponding author.

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
