# Peer review of "Hybrid Bis-Histidine Phenanthroline-Based Ligands to Lessen Aβ-Bound Cu ROS Production: An Illustration of Cu(I) Significance"

_molecules, 2021, doi:10.3390/molecules26247630_

Round 1

Reviewer 1 Report

The manuscript of Christelle Hureau and co-authors describes the synthesis of new ligands based on phenanthroline structure to capture Cu(I/II) from the amyloid-β peptide and to prevent the formation of ROS in the presence of physiological reductants. The structures of the new molecules, although on results already published, have an acceptable level of novelty. The characterization of the complexes is complete and the study on cyclic redox equilibrium between the Cu(I)(phenH) or Cu(I)(phenH)2 species is convincing.

I recommend publishing the manuscript after minor revision.

I suggest moving the synthetic scheme of ligands from Supporting info (Scheme S1) to Chapter 2.1 of the main text.

Page 3 line 106 capitalize “1,10-Phenanthroline…”

Page 4 line 116 capitalize “2-Carboxy-1,10-phenanthroline…”

Page 4 line 121 capitalize ” 2,9-Dimethyl-1,10-phenanthroline…

Reviewer 2 Report

The paper by Drommi and coworkers is a nice piece of work, well written and organized, which describes the spectroscopic and electrochemical characterization of a series of Cu(II)/Cu(I) complexes of phenantroline derivatives. Unfortunately, no practical application looks envisageable, also because no hint is given on the toxicity of both these ligands and their complexes.
The paper is certainly valuable for didactic purposes, but I'm not sure that "molecules" is the better journal to be published in. 
I don't have any suggestions to improve it. It could be either rejected (re-addressed to a different journal) or published as it is. I leave the decision to the Editor.

Reviewer 3 Report

The manuscript by Hureau and co-workers describe the preparation of phenanthroline-based ligands to prevent their copper complex dimers. This manuscript is well written and provided very useful information to understand the relationship between the activity and the structural feature. Therefore, this reviewer recommends this manuscript be published in Molecules.